# Effects of CO_2_ Concentration and the Uptake on Carbonation of Cement-Based Materials

**DOI:** 10.3390/ma15186445

**Published:** 2022-09-16

**Authors:** Qi Yu, Bingbing Guo, Changjiang Li

**Affiliations:** 1School of Civil Engineering, Qingdao University of Technology, Qingdao 266033, China; 2Qingdao Qingjian New Material Group Co., Ltd., Qingdao 266108, China; 3School of Civil Engineering, Xi’an University of Architecture and Technology, Xi’an 710055, China; 4National Key Laboratory of Green Building in West China, Xi’an University of Architecture and Technology, Xi’an 710055, China

**Keywords:** cement-based materials, carbonation reactions, CO_2_ concentration, CO_2_ uptake, thermodynamic modelling

## Abstract

Carbonation seriously deteriorates the durability of existing reinforced concrete structures. In this study, a thermodynamic model is used to investigate the carbonation reactions in cement-based materials. The effects of the concentration and amounts of CO_2_ on the carbonation behaviors of mortar are discussed. The simulation results show that the mechanisms of the carbonation reaction of cement-based materials at different CO_2_ concentrations may be different. Nearly all of the hydrate phases have a corresponding CO_2_ concentration threshold, above which the corresponding carbonation reaction can be triggered. The thresholds of the C-S-H phases with different Ca/Si ratios are different. The calculation results also show that the phase assemblages in cement paste after being completely air-carbonated, primarily consist of a low-Ca/Si ratio C-S-H, strätlingite, CaCO_3_ and CaSO_4_. The pH of the pore solution exhibits a significant decrease when a higher Ca/Si ratio C-S-H phase is completely decalcified into a lower Ca/Si ratio C-S-H phase, by increasing the CO_2_ uptake. Additionally, the experimental results and the previously published investigations are used to validate the simulation results.

## 1. Introduction

The carbonation of concrete is regarded as one of the primary factors in the diminishing durability of reinforced concrete (RC) structures [1]. Generally, the pH value of a concrete pore solution is between 12.5 and 13.5, and in such a high-alkaline environment, a protective passivation layer will be produced at the embedded reinforcing steel surface [2]. However, CO_2_ penetrates into concrete from the external environment, causing a decrease in the alkalinity of the concrete pore solution, and it has been reported that the pH value of the pore solution in carbonated concrete can be lower than 9 [3]. Undoubtedly, the protective passivation layer is easily destroyed at such a low pH value, hence causing the corrosion of the reinforcing steel to initiate easily. This can reduce the effective cross-section of the reinforcing steel and lower the bearing capacity of RC structures, and even lead to the collapse of the structure. With the rapid development of the global economy, the CO_2_ concentration in the atmosphere has been rising. It has been reported that the concentration level was at 391 ppm in 2011, and it has exceeded the pre-industrial levels by approximately 40% [4]. Thus, the degradation of the durability of RC structures caused by carbonation has always attracted much attention [5,6,7,8], and many researchers have been devoted to the investigation of carbonation behaviors of cement-based materials [9,10,11,12,13,14,15,16,17,18].

Cement-based materials consist of solid, liquid and gas phases. The solid phase includes a hardened cement paste and aggregates, and the aggregates do not participate in any chemical reactions during the carbonation process. The hydrates of ordinary Portland cement (OPC) primarily include calcium silicate hydrate (C−S−H), portlandite (CH), monosulfoaluminate (AFm) and ettringite (AFt). The liquid and gas phases are complementary, and they consist completely of the entire pore volume in cement-based materials. Large amounts of free ions exist in the liquid pore solution. These primarily include Ca^2+^, OH^−^, K^+^, Na^+^ and ${\mathrm{SO}}_{4}^{2-}$. Due to diffusion and capillary pore pressure, the CO_2_ in the atmosphere easily enters into the pore spaces of concrete and dissolves into the pore solution. It then reacts with Ca^2+^ and OH^−^, precipitating into calcite (CaCO_3_). The consumption of Ca^2+^ in the pore solution then inevitably leads to a progressive decalcification of the hydrated cement paste [2]. Thus, the carbonation behaviors of cement-based materials are assumed to be the result of a series of chemical reactions between the pore solution and the hydrated cement paste. Existing studies [11,17] indicate that the CO_2_ concentration has a significant influence on the carbonation behaviors of cement-based materials. These studies were based on experiments, and some representative CO_2_ concentrations were selected to be used on the carbonation of cement-based materials. For example, CO_2_ concentrations of 100%, 10% and 3% were used in a study by Castellote et al. [11]. CO_2_ concentrations of 100%, 50% and 0.03% were used in another study by Anstice et al. [17]. Few studies have used successive CO_2_ concentrations, especially at low concentrations that are closer to practical engineering. This is primarily because it requires a long-period and a large amount of experimental work.

From the thermodynamic point of view, the CO_2_ ingress destroys the existing thermodynamic equilibrium between the pore solution and cement hydrates. The results include the formation of calcite, a decrease in the pH value and the decalcification of the cement hydrates, which serves to achieve a new thermodynamic equilibrium in the cement-based materials [19]. Thus, thermodynamic modeling will be a powerful tool to investigate the carbonation behaviors of cement-based materials [20]. In this study, a thermodynamic model based on the dissolution equilibrium of gas in an aqueous solution and the dissolution/precipitation of the phase equilibrium between the pore solution and the cement hydrates is proposed in order to investigate the essential chemical reactions between carbonation environments and cement-based materials. The effects of the CO_2_ concentration and its uptake on the carbonation behavior of the cement-based materials are discussed. It is noted that the carbonation of cement-based materials is a typical reactive transport process, but the transport of species are not taken into account in this study. The thermodynamic simulation in this study is provided: (1) at chemical equilibrium, and (2) for a closed system.

## 2. Thermodynamic Modeling

Generally, the amount of a gas that dissolves into an aqueous solution relates to its partial pressure, which can be described by Henry’s law [21]:(1)$$m_{i}=H_{i}\cdot p_{i}$$
where *m_i_* denotes the molar concentration of the gas *i* that dissolves into an aqueous solution (mol/kg); *p_i_* is the partial pressure of the gas *i* (kPa); *H_i_* is Henry’s law constant, and Henry’s law constant of CO_2_ is 3.39 × 10^−4^ mol/kg⋅kPa at the standard atmospheric pressure and at 25 °C [21].

With the dissolution of CO_2_ into the pore solution, the dissolution/precipitation reactions between the pore solution and the cement hydrates occur in order to achieve a new thermodynamic equilibrium. The equilibrium reactions can be expressed using the law of mass action (LAM) equation [22]:(2)$$K_{p}={\prod_{i}\left(\gamma_{i}c_{i}\right)}^{n_{{}_{i},p}}$$
where *n_i,p_* is the stoichiometric coefficient of the ion *i* that participates in the dissolution/precipitation reaction; *γ_i_* and *c_i_* are the activity coefficient and the concentration of the ion *i*, respectively; *K_p_* is the thermodynamic equilibrium constant of the pure phase *p* in the cement hydrates, which can be calculated using the following equation [22]:(3)$$K_{p}=\exp(-\frac{\Delta_{r}G_{T}^{0}}{RT})$$
where *R* is the universal gas constant (8.314551 J/(mol⋅K)); *T* is the temperature; Δ*_r_G_T_*^0^ is the standard Gibbs energy of the reaction. The CEMDATA database has already become more developed and can give *K_p_* for nearly all of the phases of a cement hydrate. This makes the thermodynamic calculations more reliable, and CEMDATA18 was used in this study.

Additionally, the ionic activity coefficient relates to the ionic strength, which can be described using Davies equation or the extended Debye–Hückel equation [22].
(4)$$\log(\gamma_{i})=-A\cdot z_{i}^{2}\cdot (\frac{\sqrt{\mu }}{1+1.4\sqrt{\mu }}-0.3\mu )$$
or,
(5)$$\log(\gamma_{i})=\frac{-Az_{i}^{2}\cdot \sqrt{\mu }}{1+B\cdot a_{i}\cdot \sqrt{\mu }}+b_{i}\cdot \mu$$
where *z_i_* is the charge of the ion *i*; *A* and *B* are constants, and they are determined by the temperature; *a_i_* and *b_i_* are ion-specific parameters from the mean-salt activity-coefficient data; *μ* is the ionic strength of the aqueous solution, which is defined as the following equation:(6)$$\mu =0.5\sum_{i=1}^{N}z_{i}^{2}c_{i}$$

The carbonation reactions of the cement-based materials at different CO_2_ volume concentrations and CO_2_ amounts were solved by using the PHREEQC COM module (Version 3.0, United States Geological Survey, Reston, VA, USA) in MATLAB (2016a, MathWorks. Inc., Natick, MA, USA) [23,24].

## 3. Experimental Procedures

An experiment was conducted to verify the proposed thermodynamic model. Ordinary Portland cement PO42.5, produced by Yatai Group (Harbin, China), was used, and its chemical composition contained CaO (62.24%), SiO_2_ (20.89%), Al_2_O_3_ (5.44%), Fe_2_O_3_ (3.96%), MgO (1.71%), SO_3_ (2.65%), CO_2_ (0.48%), K_2_O (1.07%), Na_2_O (0.12%) and LOI (1.44%). In addition, river sand and a polycarboxylate based plasticizer were used. The contents of the water, cement, sand and water reducer were 315.79 kg/m^3^, 631.58 kg/m^3^, 1736.84 kg/m^3^ and 0.751 kg/m^3^, respectively. Following the 28-day curing, the compressive strengths of the cubic specimens (70.7 × 70.7 × 70.7 mm^3^) were measured, and were consistently approximately 36 MPa.

A schematic of the experimental procedure is illustrated in Figure 1. The cubic mortar specimens with a size of 400 × 100 × 100 mm^3^ were prepared, and all of the specimens were cured in a saturated calcium hydroxide solution for six months. Then, the cubic specimens were cut into thin slices with a thickness of approximately 3 mm in order to shorten the next experimental period. In addition, the top and bottom surfaces were omitted. Seven slices were selected, and their initial weights were measured using a high precision balance with 0.1 mg. Then, these seven thin specimens were dried in a vacuum drying oven at 50 °C. The weights of the thin specimens were recorded every seven days during the drying process, and when the difference between two continuous recorded weights was less than 0.1%, the specimens were assumed to be at a constant weight. Specifically, this indicated that they did not contain any capillary pore water. Next, the specimens were placed into a closed carbonation chamber at 20 °C where the CO_2_ volume concentration and the relative humidity were 20% and 70%, respectively. To make these seven slices completely carbonated, the carbonation period was set at 60 days. At the end of the carbonation, a 1 wt% phenolphthalein aqueous solution was sprayed on the fracture surface of a thin specimen, and it was found that the color did not change. Then, these specimens were immediately again dried to a constant weight, and their weights were again recorded. Based on the reaction (CO_2_ + 2OH^−^ → ${\mathrm{CO}}_{3}^{2-}$+ H_2_O) and the variation in the weights of the specimens, the CO_2_ moles (*N*_CO2_) that reacted with the specimens could be calculated using the following equation:(7)$$N_{{\mathrm{CO}}_{2}}=\frac{w\cdot \theta_{m}}{M_{{\mathrm{CO}}_{2}}-M_{H_{2}O}}$$
where *w* is the increase in the weights of the mortar before and after carbonation; $w=\frac{m_{2}-m_{1}}{m_{1}}$ (*m*_1_ and *m*_2_ are the weights of the fully dried specimens before and after the carbonation, respectively, which can be also seen in Figure 1); θ_m_ is the density of the mortar, and the value was 2.685 kg/L; and M_CO2_ and M_H2O_ are the molar masses of CO_2_ and H_2_O, respectively.

The thin specimens were then saturated with distilled water using a vacuum water saturation instrument for 24 h. In order to achieve the equilibrium of the interaction between the pressed distilled water and the cement hydrates, the saturated specimens were sealed using plastic membranes for seven days. Then, the pore solutions of the thin specimens were extracted via mechanical pressure. Following the extraction, the pH value and the ionic concentration in the pore solution were diluted, and then immediately measured using a pH meter with a high precision and ion chromatography, respectively. The measured values were used to compare with the simulation results.

In addition, a carbonated thin specimen was ground into a fine powder for the thermogravimetric analysis (TGA) and X-ray diffraction (XRD) measurements. In the TGA measurements, a Mettler Toledo TGA/SDTA 851e instrument (Mettler-Toledo Instruments (Shanghai) Co., Ltd, Shanghai, China) was used, and approximately 13 mg of the powder was heated at 10 °C/min from 20 to 1000 °C, and nitrogen was used to purge. The XRD measurements were conducted using an X’ Pert PRO diffractometer with Cu-Kα radiation (λ = 0.15419 nm), which was from PANalytical B.V. (Malvern Panalytical Ltd., Malvern) in United Kingdom, and a 2θ range of 10−90° was scanned for 45 min.

## 4. Model Parameters

The initial ionic concentrations in the pore solution and the initial amounts of the hydrate phases needed to be input into the thermodynamic model to calculate the chemical reactions between the carbonation environments and mortar. The initial ionic concentrations of K^+^, Na^+^, Ca^2+^, ${\mathrm{SO}}_{4}^{2-}$ and OH^−^ in the pore solution were measured during the experimental process according to the method referred to in Section 3, and the values are given in Table 1. The initial hydrate assemblage and their contents can be obtained using the thermodynamic modelling of the hydration of the Portland cement, provided by the chemical composition of the cement, content of the cement, water to cement ratio and the degree of hydration [25]. Considering that the mortar had been cured for six months before the carbonation, it was assumed that the cement was close to a complete hydration. Then, based on the contents of the cement and water of the fabricated mortar specimen given in the last section, the calculation results of the thermodynamic model showed that the hydrate assemblage of the cement consisted of C-S-H (3CaO⋅2SiO_2_⋅5H_2_O, 2.196 mol/L of mortar), portlandite (2.115 mol/L of mortar), AFt (0.048 mol/L of mortar), AFm (0.132 mol/L of mortar) and monocarbonate (0.157 mol/L of mortar).

## 5. Results and Discussion

### 5.1. Carbonation Behaviors at Different CO_2_ Volume Concentrations

Assuming that the CO_2_ amount was sufficient, the effects of the CO_2_ concentrations on the carbonation behaviors of the cement-based materials are discussed in the context of the thermodynamic model in this section. Figure 2 plots the hydrate assemblages of the completely carbonated mortar as a function of the CO_2_ concentrations. The CO_2_ gas cannot dissolve into any aqueous solution when the CO_2_ concentration is too low, and thus, the cement-based materials do not undergo any carbonation reactions when the CO_2_ volume concentration is lower than 1.95 × 10^−15^ (Figure 2). When a sufficient CO_2_ amount is provided, the CO_2_ concentration that can trigger the AFm carbonation is the lowest among all of the cement hydrates, and it is followed by the CH, C-S-H and AFt. In addition, apart from the C-S-H phase with a Ca/Si ratio = 1.0, every hydrate phase has the corresponding CO_2_ concentration threshold, above which the corresponding carbonation reaction can be triggered. These values, predicted by the thermodynamic model, are 1.95 × 10^−15^ for the AFm (Figure 2a), 1.0 × 10^−13^ for the CH (Figure 2c) and 5.50 × 10^−8^ for the AFt (Figure 2a). In addition, the C-S-H phases with different Ca/Si ratios have different CO_2_ concentration thresholds, such as 1.91 × 10^−12^ (C-S-H with Ca/Si ratio = 1.5) and 1.35 × 10^−9^ (C-S-H with Ca/Si ratio = 1.0). Overall, the effect of the CO_2_ concentration on the carbonation reactions of CaSO_4_ and the low-Ca/Si ratio C-S-H phases is less significant than that of the AFm, AFt, CH, monocarbonate, strätlingite and the high-Ca/Si ratio C-S-H phases. Moreover, the CO_2_ volume concentration in the current atmosphere is approximately 3.8 × 10^−4^ (which has been remarked in Figure 2), and this means that most of the hydrate phases will be completely carbonated if sufficient CO_2_ penetrates into the cement-based materials.

The simulation results (Figure 2a) indicate that the AFm carbonation leads to the improvements in the AFt and monocarbonate contents. However, with an increase in the CO_2_ volume concentration, the monocarbonate can also be carbonated (see Figure 2a), and simultaneously, the first decalcification of the C-S-H phase occurs (see Figure 2b). Subsequently, the AFt phase undergoes carbonation accompanied by the formation of CaSO_4_ (see Figure 2a). However, with the continuously increasing CO_2_ concentration, CaSO_4_ is also gradually carbonated. In addition, the CaCO_3_ content increases when any hydrate phase, apart from the AFm (i.e., CH, C-S-H, AFt, monocarbonate and CaSO_4_), is carbonated, which can be seen in Figure 2. This calculation result is in line with the well-known conclusion that CaCO_3_ is the most primary product of the carbonation reactions of the cement hydrates. Figure 2b illustrates the contents of the C-S-H phases with different Ca/Si ratios at different CO_2_ volume concentrations, and it is clear that the C-S-H has two different decalcification behaviors with the increasing CO_2_ concentration, which means that the rise in the CO_2_ concentration can cause a completely carbonated C-S-H to possibly experience three stages. In the first stage, it cannot induce the decalcification of the C-S-H due to the very low CO_2_ concentration. When the CO_2_ volume concentration is between 1.91 × 10^−12^ and 1.35 × 10^−9^, the C-S-H phase with a Ca/Si ratio = 1.5 is completely decalcified into that of a Ca/Si ratio = 1.0. When the CO_2_ volume concentration is above 1.35 × 10^−9^, the C-S-H is completely decalcified into that of a Ca/Si ratio = 2/3, which is the third stage. Castellote et al. [17] employed ^29^Si Magic Angle Spinning-Nuclear Magnetic Resonance (^29^Si MAS NMR), TG and XRD to investigate the chemical changes and the phase analysis of OPC pastes carbonated at different CO_2_ concentrations. They found that the Ca/Si ratio of the C-S-H phase became lower with the increasing CO_2_ concentration during the carbonation process. This agrees well with the simulation results of the proposed thermodynamic model. The simulation results in Figure 2b also indicate that the CO_2_ concentration in the current atmosphere can hardly cause a carbonation reaction of the C-S-H phase with a lower Ca/Si ratio, for example, 2CaO⋅3SiO_2_⋅4.5H_2_O. However, when the CO_2_ concentration is higher than 1.62%, the C-S-H phase is completely carbonated. The experimental results reported by Castellote et al. [17] also indicate that the C-S-H phase completely disappears when the CO_2_ concentration is 10% and 100%. In addition, the decalcification of the C-S-H phase caused by the carbonation results in the formation of strätlingite, has also been reported by Shi et al. [26]. Figure 2c presents the variations in the CH and CaCO_3_ contents in cement hydrates as a function of the CO_2_ concentration. The CH cannot be carbonated when the CO_2_ concentration is lower than 1.0 × 10^−13^. In addition, it can be clearly observed that the CaCO_3_ content in the completely carbonated cement-based materials as a main carbonation product is closely related to the CO_2_ concentration.

The thermodynamic model predicts the pH in the pore solution as a function of the CO_2_ volume concentration, as illustrated in Figure 3. With an increase in the CO_2_ volume concentrations, the changing curve of the pH value in the pore solution can be divided into two stages: in the first stage, because the CO_2_ concentration is very low, it does not cause any carbonation reactions in the cement-based material, or only the AFm and CH phases are carbonated, and the pH value of the pore solution has made little change and is above 13.0. In the second stage, the pH value of the pore solution is susceptible to the effect of the CO_2_ concentration. When the decalcification of a high-Ca/Si ratio C-S-H occurs, the pH value begins to decrease. Some investigations have indicated that the pH value of the concrete pore solution that can cause a transition from passive to active corrosion of the embedded reinforcing steel is between 10 and 9.4, and this value may be higher in the presence of chloride ions [6,27]. By combining this with the results in Figure 3, it is concluded that the carbonation cannot induce the corrosion of the reinforcing steel when the CO_2_ volume concentration is below 3.39 × 10^−8^. However, the CO_2_ concentration in the current atmosphere is about 3.8 × 10^−4^, and this means that the current atmosphere is capable of inducing the corrosion of the reinforcing steel.

Grove et al. [28] studied the structural changes in hardened C_3_S cement pastes due to the carbonation using the combination of ^29^Simagic angle spinning nuclear magnetic resonance spectroscopy (^29^Si NMR) and analytical transmission electron microscopy (TEM). They found that the structures of the CO_2_-carbonated and air-carbonated cement pastes are different. This agrees with the results of Figure 2 and Figure 3 that the carbonation reaction mechanisms of the cement-based materials at the different CO_2_ concentrations are different.

The above results indicate that the carbonation reactions at different CO_2_ concentrations are different, and since ${\mathrm{CO}}_{3}^{2-}$, ${\mathrm{SO}}_{4}^{2-}$ and Ca^2+,^ in the pore solution, participate in the carbonation reactions of the hydrate phases, the ionic concentrations of the pore solution at different CO_2_ concentrations are determined by the corresponding carbonation reactions. Figure 4 plots the ionic concentrations of ${\mathrm{CO}}_{3}^{2-}$, ${\mathrm{SO}}_{4}^{2-}$, Ca^2+^, K^+^ and Na^+,^ in the pore solution, as a function of the CO_2_ volume concentration predicted by the thermodynamic model. The ${\mathrm{SO}}_{4}^{2-}$ concentration has a significant increase when CaSO_4_ undergoes a carbonation reaction with an increase in the CO_2_ concentration. The ${\mathrm{CO}}_{3}^{2-}$ concentration remains very low and is barely affected by the CO_2_ concentration. This is because it is very difficult for CO_2_ to dissolve into an aqueous solution when the CO_2_ concentration is low, and with the increasing CO_2_ concentration, ${\mathrm{CO}}_{3}^{2-}$ produced due to the CO_2_ dissolution participates in the carbonation reactions, existing as CaCO_3_ or a monocarbonate. The Ca^2+^ concentration displays an increase only when the AFt is carbonated with the increasing CO_2_ concentration. In addition, Na^+^ and K^+^ do not participate in any carbonation reactions, and some carbonation reactions can increase the amount of water in the pore solution from the insight of the macroscopic chemical reactions, e.g., CO_2_ + Ca(OH)_2_ → CaCO_3_ + H_2_O, and there are no species transports in a closed system. Thus, the Na^+^ and K^+^ concentrations show a decrease (Figure 4b), in particular when the carbonation reaction of a hydrate phase is triggered.

### 5.2. Carbonation Behaviors with a Different CO_2_ Uptake

When mortar is exposed to natural carbonation (i.e., the CO_2_ volume concentration is 3.8 × 10^−4^), the phase assemblages as a function of the CO_2_ amount are predicted using the thermodynamic model, as illustrated in Figure 5. When the CO_2_ amount increases, the AFm and CH phases are carbonated first, followed by the C-S-H and AFt phases, which is similar to the results from the different CO_2_ concentrations. The AFm carbonation improves the AFt and monocarbonate contents (Figure 5a). The carbonation of the C-S-H lowers its Ca/Si ratio (Figure 5c), leads to the formation of strätlingite (Figure 5a) and improves the content of CaCO_3_ (Figure 5b), and the monocarbonate is carbonated along with the C-S-H. Shi et al. [26] has also reported similar results. Furthermore, the simulation results shown in Figure 5 indicate that the phase assemblages in the cement paste after being completely air-carbonated, primarily consist of a low-Ca/Si ratio C-S-H, strätlingite, CaCO_3_ and CaSO_4_.

The thermodynamic model predicts the pH in a pore solution of the air-carbonated mortar as a function of the CO_2_ amount, as illustrated in Figure 6. With an increase in the CO_2_ amount, the pH has a significant decrease only when a higher Ca/Si ratio C-S-H phase is completely decalcified into a lower Ca/Si ratio C-S-H phase. However, it displayed little change during the carbonation process of each phase. There are three total significant decreases, and the first decrease (to pH = 11.08) occurs after the complete carbonation of 3CaO⋅2SiO_2_⋅5H_2_O, and the second decrease (to pH = 9.84) occurs after the complete carbonation of CaO⋅SiO_2_⋅2H_2_O, and the third decrease (to pH = 7.77) occurs after the complete carbonation of 2CaO⋅3SiO_2_⋅4.5H_2_O.

### 5.3. Model Validation and Experimental Results

Figure 7 shows the weights of all of the fully dried specimens in the experiment before and after carbonation. The increase in the weight of the specimens caused by the carbonation varies between 5.04% and 5.44%, and the average value is 5.18%. The discreteness of the experimental result is negligible. According to Equation (7), the CO_2_ moles (*N*_CO2_) that react with each liter of mortar can be calculated, and the value is 5.349 mol. The CO_2_ volume concentration in the experiment is 20%. Then, the CO_2_ moles and volume concentration are input into the thermodynamic model. The carbonation reactions can then be calculated using the proposed thermodynamic model. The calculation results include the hydrate phase assemblages and the ionic concentration of the pore solution, which are listed in Table 2 and Table 3. The results show that the hardened mortar that is completely carbonated only contains a low-Ca/Si ratio C-S-H (2CaO⋅3SiO_2_⋅4.5H_2_O), CaCO_3_, CaSO_4_ and strätlingite, and the other phases are completely carbonated.

Figure 8 plots the first derivative (DTG) of the mass losses measured using the TGA for the carbonated mortar sample and its XRD patterns. There may be three primary peaks in the DTG cure for a carbonated mortar, that is, the C-S-H/AFt/AFm phases (30–300 °C), portlandite (400–500 °C) and CaCO_3_ (500–800 °C). However, it is very clear from Figure 8a that there is only a CaCO_3_ DTG peak. In addition, the peaks for the CH, AFt, AFm, monocarbonate and CaCO_3_ should be observed in the XRD pattern of the normal cement paste, but it can be seen in Figure 8b that there are only the peaks of CaCO_3_ and SiO_2_ in the XRD patterns of the completely carbonated mortar, which means that all of the CH, AFm and AFt are carbonated. Taken together, the XRD and TGA results indicate that the CH, AFm and AFt phases are completely reacted in a completely carbonated cement-based material, which agrees with the simulation results of the thermodynamic model (Table 2). To further validate the proposed thermodynamic model, Table 3 presents a comparison between the simulation and experimental results of the ionic concentration in the pore solution. It can be clearly seen that the simulation results have a good agreement with the experimental results.

In practical engineering, the carbonation of concrete is a reaction process from the surface to the inside, which is a typical reactive transport. It can be assumed to consist of two modules: reaction module and transport module, but they affect each other. The former is the reactions among the gas phase (CO_2_), liquid phase (concrete pore solution) and the solid phase (cement hydrates), which can be assumed in a closed system. The latter involves gas, ionic and water transports. The transport drives the system out of chemical equilibrium, and simultaneously, the chemical reactions change the diving force of the transport, e.g., the concentration gradient. Consequently, the reactive transport code is required for simulating the concrete carbonation in practice.

## 6. Conclusions

In this study, the thermodynamic model was proposed in order to investigate the carbonation reactions in cement-based materials. The effects of the CO_2_ concentration and uptake on the carbonation behaviors of cement-based materials were investigated. The experimental results in this study and other published investigations [17,26,28] validated the accuracy of the proposed thermodynamic model.

The calculation results show that nearly all of the hydrate phases have a corresponding CO_2_ concentration threshold, above which the corresponding carbonation reaction can be triggered. The threshold value are 1.95 × 10^−15^ for the AFm, 1.0 × 10^−13^ for the CH and 5.50 × 10^−8^ for the AFt, respectively. In addition, the C-S-H phases with different Ca/Si ratios have different threshold values. Therefore, the carbonation reaction mechanisms of the cement-based materials at different CO_2_ concentrations are different. The phase assemblages in the cement paste, after being completely air-carbonated, primarily consist of a C-S-H with a low Ca/Si ratio, strätlingite, CaCO_3_ and CaSO_4_. It is found that the CO_2_ concentration in the current atmosphere is capable of causing carbonation reactions of most of hydrate phases. The pH of the pore solution shows a significant decrease when a higher Ca/Si ratio C-S-H phase is completely decalcified into a lower Ca/Si ratio C-S-H phase by increasing the CO_2_ amount.

## Figures and Tables

**Figure 1 materials-15-06445-f001:**
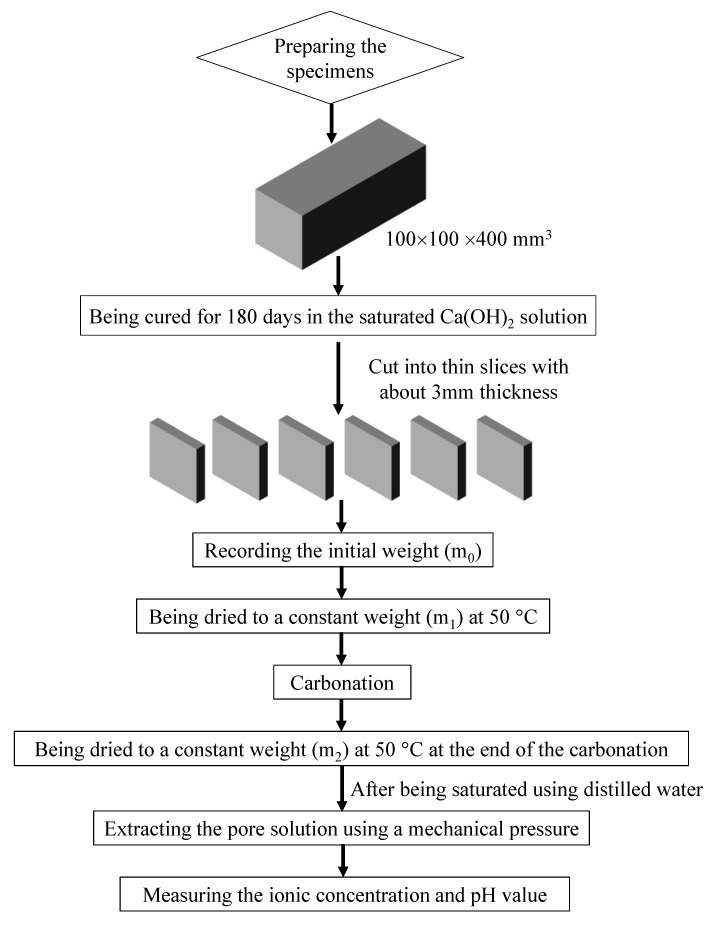
Schematic of the experimental procedure.

**Figure 2 materials-15-06445-f002:**
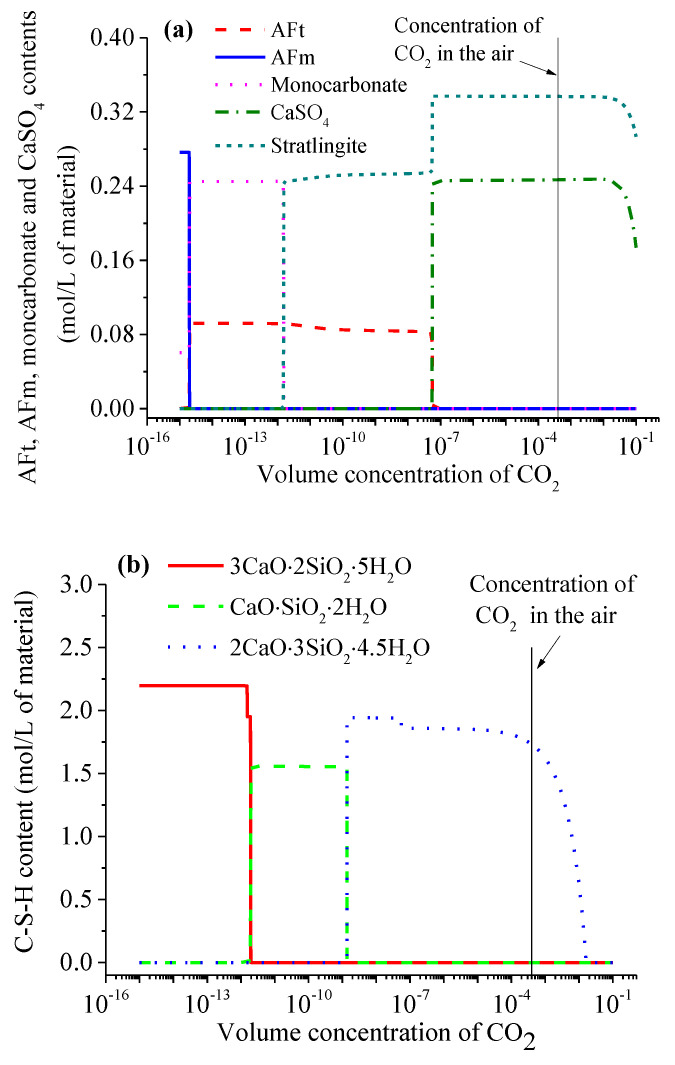
Phase assemblages of the hardened cement paste in a completely carbonated mortar as a function of the CO_2_ concentrations: (**a**) AFt, AFm, monocarbonate and CaSO_4_ phases; (**b**) C-S-H phases with different Ca/Si ratios; (**c**) CH and CaCO_3_ phases.

**Figure 3 materials-15-06445-f003:**
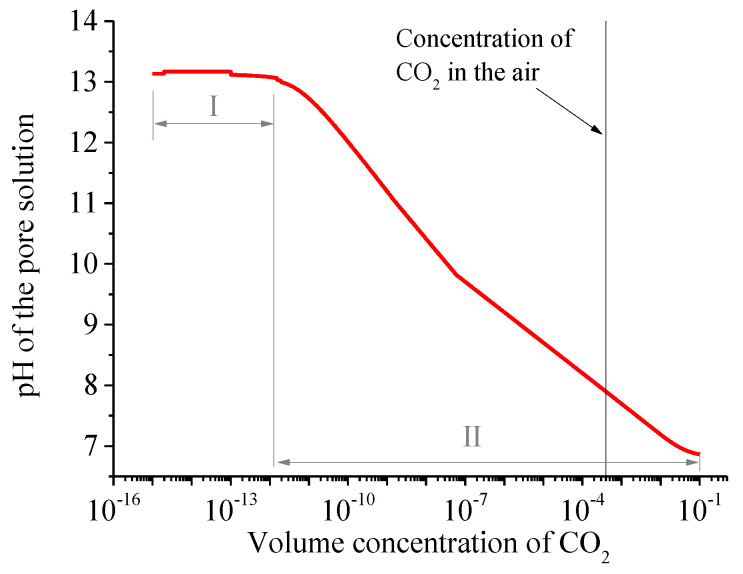
pH in the pore solution of the completely carbonated mortar as a function of the CO_2_ concentrations. Ι and ΙΙ in the figure represent the first and second stages, respectively.

**Figure 4 materials-15-06445-f004:**
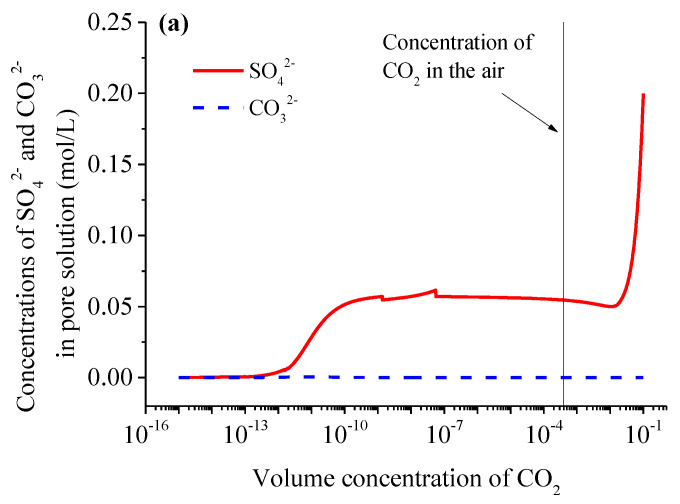
Ionic concentrations in the pore solution of the completely carbonated mortar as a function of the CO_2_ concentrations: (**a**) and (**b**) represent the concentrations of anions and cations, respectively.

**Figure 5 materials-15-06445-f005:**
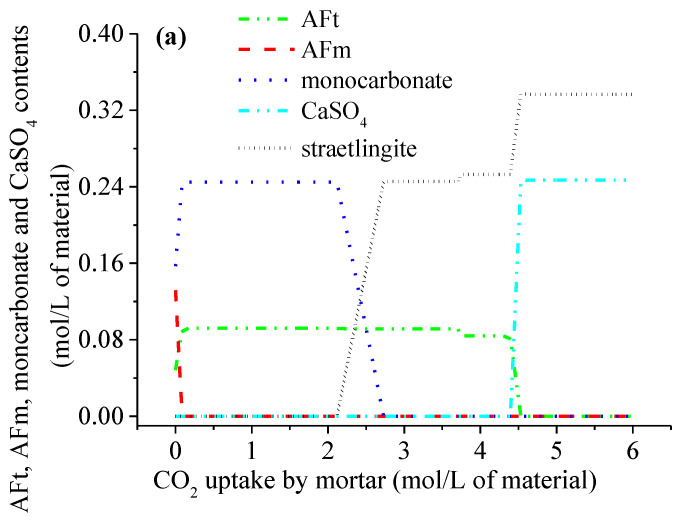
Phase assemblages of the hardened cement paste in the air-carbonated mortar as a function of the CO_2_ uptake: (**a**) AFm, AFt, moncarbonate and CaSO_4_; (**b**) CH and CaCO_3_; (**c**) C-S-H.

**Figure 6 materials-15-06445-f006:**
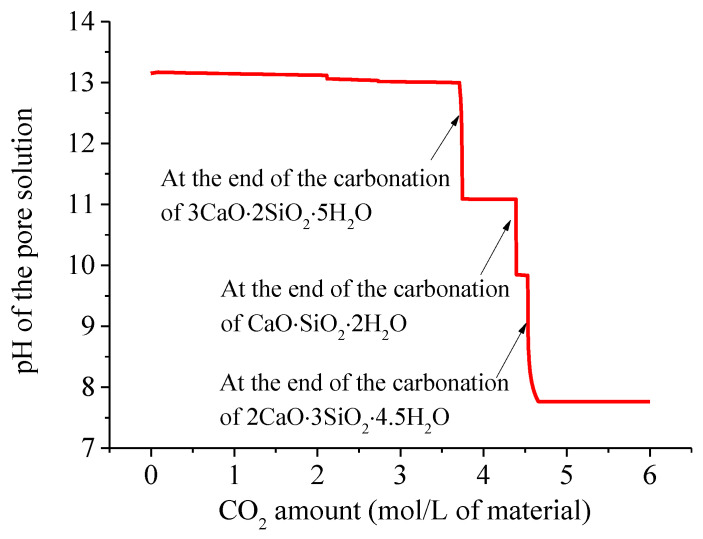
pH in the pore solution in air-carbonated mortar as a function of CO_2_ uptake.

**Figure 7 materials-15-06445-f007:**
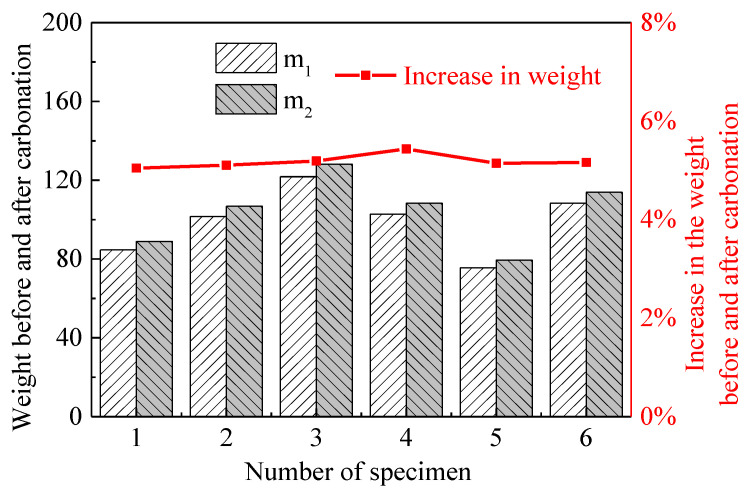
Weights of the dried specimens before and after the carbonation.

**Figure 8 materials-15-06445-f008:**
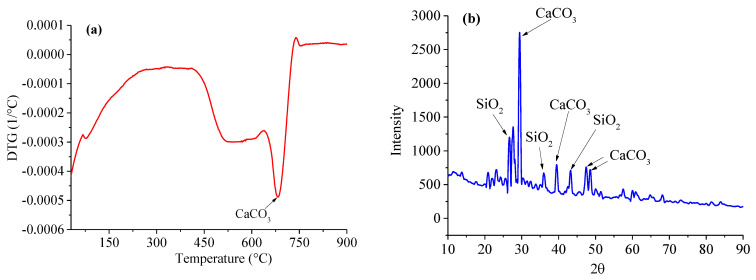
(**a**) DTG curves and (**b**) the XRD patterns of the carbonated mortar sample.

**Table 1 materials-15-06445-t001:** Initial concentrations of the free ions and the pH value in the pore solution of the model.

Ion	Na^+^ (mM)	K^+^ (mM)	Ca^2+^ (mM)	${\mathrm{SO}}_{4}^{2-}$ (mM)	pH
Concentration	79.0	65.0	0.4	1.43	13.14

**Table 2 materials-15-06445-t002:** Simulation results of the phase composition for the mortar that is completely carbonated.

Hydrate Phase	C-S-H (2CaO⋅3SiO_2_⋅4.5H_2_O)	AFt	AFm	CH	Monocarbonate	CaCO_3_	CaSO_4_	Strätlingite
Amount (mol/L of mortar)	1.668	0	0	0	0	4.814	0.247	0.337

**Table 3 materials-15-06445-t003:** Comparison between the experimental and simulation results of the ionic concentration.

Ion	Na^+^ (mM)	K^+^ (mM)	Ca^2+^ (mM)	${SO}_{4}^{2-}$ (mM)	pH
Experimental results	63.97	46.01	6.34	55.88	7.93
Simulation results	49.75	39.83	8.05	54.01	7.77

## Data Availability

Not applicable.

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
