# Peer review of "Effects of CO2 Concentration and the Uptake on Carbonation of Cement-Based Materials"

_materials, 2022, doi:10.3390/ma15186445_

Round 1

Reviewer 1 Report

The paper presents a numerical and experimental study on carbonation of cement-based materials. Results obtained by calculations of chemical reactions at equilibrium with Phreeqc are compared to experimental data of totally carbonated materials. Overall, the paper is well written and relatively well documented. However, the results presented do generally not show much novelty w.r.t. existing studies in literature. Besides, the authors should mention clearly and discuss that their numerical results are provided: (1) at chemical equilibrium, and (2) for a closed system. Regarding the second point, in reality the cementitious materials subjected to carbonation in unsaturated conditions experience diffusive ionic phenomena in conjunction with evolution of saturation degree due to drying and release of water following carbonation reactions. This means that the concentrations of ionic species and water content may be very different for points situated at different depths from the exposure surface, which affects greatly the chemical reactions. The results presented are nevertheless globally of interest; the reviewer recommends that the paper could be published after the authors respond to the comments above and the complementary remarks listed below:

-in section 2, the authors explain they use a thermodynamic model to simulate the evolution of the hydrated products upon carbonation, however they cite the code used (Phreeqc) only a few pages later: this information should be given before.

-l158-159: more information should be given regarding the initial phase assemblage calculation.

-l179: it is stated that ‘AFm is the first phase that is carbonated’. However, in the literature CH is commonly accepted as the first phase destabilized by CO2. The authors should comment on this point.

-CSH decalcification/carbonation is apparently modelled with 3 phases having different Ca/Si ratio. This is a well known method but should be much more clearly stated and explained in the paper.

-l201-202, it is stated that ‘This result validates that CaCO3 is the most primary product of carbonation reactions of cement hydrates.’ This a very well known result…

-caption of Fig. 2 is unclear and incorrect; in particular, Fig. 2c) is apparently not referenced nor explained in the text.

-l263-266: it is stated that ‘carbonation reactions can increase the amount of water in the pore solution. Thus, the Na+ and K+ concentrations show a decrease…’ However and as mentioned above, carbonation is generally associated with a drying process that leads to homogenize (and decrease) the saturation degree. Moreover, a change of Na+ and K+ concentrations locally necessarily triggers diffusive phenomena that also tend to homogenize these concentrations. To conclude, the reviewer does not agree with these statements, which have to be mitigated by recalling that the results presented have some limitation related to the hypothesis of closed system.

-the x-axis label (‘CO2 amount of carbonating mortar’) of Fig. 5 is unclear. What do the authors mean with carbonating mortar? Moreover, the quantity ‘CO2 amount’ is not very comprehensive to the reviewer’s view. Is it the CO2 that has reacted? It cannot easily be compared to experimental data which rather present profiles of mineral assemblage as a function of the depth from the exposure surface. From this viewpoint, it is a little surprising that the authors do not mention as a perspective that applying a reactive transport code would lead to much more accurate results…

-in Fig. 7, the y-axis label is written with the term ‘carbonization’.

Reviewer 2 Report

The authors present quite interesting and original studies on the effect of the concentration and amount of CO2 on the carbonation of cement-based materials studied using a thermodynamic model. These studies are relevant and have novelty. However, there are some comments on this manuscript:

 1) For the cement used in the experiments, it is necessary to specify its brand, manufacturer, city and country. (line 103)

2) In the given chemical composition of the cement used, when summing up the values, 98.56% is obtained than the missing 1.44% presented?

3)It is necessary to cite the manufacturer's city and country for the equipment used during the study.

4) Captions to figures should be made in 9pt font. With a certain indentation from the main text.

 5)In Figure 6, CO2 is not fully displayed in the area of the axis signature. It needs to be fixed.

6) In Figure 8 (a), the temperature scale is not quite correct after 750 degrees it goes 90, probably there should be 900?

7) How the compounds shown in Figure 8 (b) were identified, it is necessary to supplement this information in the methodology. Based on the X-ray there are only compounds CaCO3 and SiO2 ... and nothing else?

8) In the list of references for the 18th position, it is necessary to give the name and surname of the authors.

Reviewer 3 Report

The manuscript “Effects of concentration and amount of CO2 on the carbonation of cement-based materials studied by a thermodynamic model” is very good.  The title must be shortened. The research is interesting. Authors must research new research. Compare research with recent results. The abstract clearly presented the research, but the conclusion was not linked to the abstract. Fix the conclusion. Explain why there is a 28 day wait. Are the comparisons made with the same sample geometry.Accept after minor revision.

Round 2

Reviewer 1 Report

The authors have nicely responded to the reviewer comments, and have provided justifications and clarified some aspects in the revised version of the manuscript; its quality has been greatly improved. The paper deserves now to be published, in the opinion of the reviewer.